# Conservation of Imprinting and Methylation of *MKRN3*, *MAGEL2* and *NDN* Genes in Cattle

**DOI:** 10.3390/ani11071985

**Published:** 2021-07-02

**Authors:** Junliang Li, Weina Chen, Dongjie Li, Shukai Gu, Xiaoqian Liu, Yanqiu Dong, Lanjie Jin, Cui Zhang, Shijie Li

**Affiliations:** 1College of Life Science, Agricultural University of Hebei, Baoding 071000, China; lijunliangyx@163.com (J.L.); gushukai666@163.com (S.G.); liuxiaoqian121@163.com (X.L.); dongyanqiu981103@163.com (Y.D.); Jin15833429940@163.com (L.J.); 2Department of Traditional Chinese Medicine, Hebei University, Baoding 071000, China; Chenweina429@163.com; 3College of Bioscience and Bioengineering, Hebei University of Science and Technology, Shijiazhuang 050081, China; ldj618@163.com

**Keywords:** cow, *MKRN3* gene, *MAGEL2* gene, *NDN* gene, imprinting, DNA methylation, DMR

## Abstract

**Simple Summary:**

In mammals, imprinted genes play key roles in embryonic growth and postnatal behavior and their misregulation has been associated with several developmental syndrome and cancers. At present, more than 150 imprinted genes have been identified in humans and mice. Cattle is an economically important farm animal, but compared to humans and mice, few genes have been reported as imprinted in this species. *MKRN3*, *MAGEL2* and *NDN* are three maternally imprinted genes in the human Prader-Willi and Angelman syndromes imprinted locus at 15q11-q13. In this study, we determined that the bovine *MKRN3*, *MAGEL2* and *NDN* genes are three paternally expressed gene, and their expression is regulated by the DNA methylation. This work could advance the genomic imprinting by increasing the knowledge of imprinted genes in bovine and also facilitate future studies to detect the physiological roles of *MKRN3*, *MAGEL2* and *NDN* genes.

**Abstract:**

Genomic imprinting is the epigenetic mechanism of transcriptional regulation that involves differential DNA methylation modification. Comparative analysis of imprinted genes between species can help us to investigate the biological significance and regulatory mechanisms of genomic imprinting. *MKRN3*, *MAGEL2* and *NDN* are three maternally imprinted genes identified in the human PWS/AS imprinted locus. This study aimed to assess the allelic expression of *MKRN3*, *MAGEL2* and *NDN* and to examine the differentially methylated regions (DMRs) of bovine PWS/AS imprinted domains. An expressed single-nucleotide polymorphism (SNP)-based approach was used to investigate the allelic expression of *MKRN3*, *MAGEL2* and *NDN* genes in bovine adult tissues and placenta. Consistent with the expression in humans and mice, we found that the *MKRN3*, *MAGEL2* and *NDN* genes exhibit monoallelic expression in bovine somatic tissues and the paternal allele expressed in the bovine placenta. Three DMRs, PWS-IC, *MKRN3* and *NDN* DMR, were identified in the bovine *PWS/AS* imprinted region by analysis of the DNA methylation status in bovine tissues using the bisulfite sequencing method and were located in the promoter and exon 1 of the *SNRPN* gene, *NDN* promoter and 5’ untranslated region (5’UTR) of *MKRN3* gene, respectively. The PWS-IC DMR is a primary DMR inherited from the male or female gamete, but *NDN* and *MKRN3* DMR are secondary DMRs that occurred after fertilization by examining the methylation status in gametes.

## 1. Introduction

Genomic imprinting is a phenomenon in which a subset of mammalian genes is completely or preferentially expressed from maternally or paternally inherited alleles. Over 100 imprinted genes cluster in 20–25 chromosomal regions in the human and mouse genomes and include both protein-coding and noncoding RNA genes in each imprinted cluster [1]. Compared with humans and mice, fewer imprinted genes have been identified in cattle. Imprinted genes play key roles in both pre- and postnatal development. In humans, aberrant imprinting is responsible for certain congenital syndromes [2], while perturbed imprinting is involved in cancers [3].

Necdin (*NDN*), melanoma antigen-like gene 2 (*MAGEL2*) and makorin ring finger protein 3 (*MKRN3*) are three paternally expressed genes located at the proximal end of the PWS/AS imprinted region, one of the best-studied imprinted clusters on human chromosome 15q11-q13. This imprinted cluster is associated with two human neurobehavioral disorders, Prader-Willi syndrome (PWS) and Angelman syndrome (AS). PWS is thought to arise from the loss of multiple paternal gene expression within the segment from *MKRN3* to the small nucleolar RNA (snoRNA) genes [4,5]. In contrast, loss of the maternally inherited copy of the *UBE3A* gene (E6-AP) leads to AS [6,7].

The cis-acting element regulating monoallelic gene expression in imprinted clusters is known as imprinting control regions (ICRs). The differential DNA methylation regions (DMRs) at CpG-rich ICRs are inherited from germline DMRs (gDMRs), which typically regulate the monoallelic expression of imprinted genes [8,9]. Apart from gDMRs, additional second DMRs are subsequently acquired after the implantation stages in response to the gDMR, which are also important for the imprinted expression of a certain gene in the cluster [10]. There are three important maternally methylated DMRs in the human PWS/AS imprinted region: a germline DMR of the PWS imprinting center (PWS-IC) [11] and somatic *NDN* [12] and *MKRN3* DMRs [13]. PWS-IC serves as the master regulator of imprinting of the PWS/AS region and hierarchically controls *NDN* and *MKRN3* DMRs in somatic tissues [14,15]. The homologous region of the human PWS/AS domain is on mouse chromosome 7C [10]. The regulatory mechanisms of genomic imprinting and genes hosted in the PWS/AS region are exquisitely maintained between mice and humans. Comparative genomics is an important way to identify functional DNA elements to regulate the expression of imprinted gene in a locus [16]. Here, we were interested in investigating the imprinting status of the cow *NDN*, *MAGEL2* and *MKRN3* genes and in identifying the three maternally methylated DMRs in the bovine *PWS/AS* imprinted region.

## 2. Materials and Methods

### 2.1. Tissue and Placental Sample Collection

Tissue samples, including heart, liver, spleen, lung, kidney, muscle, fat and brain, of 34 female Holstein cows were collected from a local abattoir. A total of 30 placental tissues of Holstein cows were collected immediately after spontaneous delivery from local cattle farm. For each placenta, the corresponding maternal whole blood and paternal sperm from five bulls used for artificial insemination were also collected to identify the parental genotype. All samples were immediately frozen in liquid nitrogen after collection for further DNA or RNA extraction. The procedures of using animals were approved by the Agriculture Research Animal Care Committee of Hebei Agriculture University. All animal experiments of this study were approved by the Agriculture Research Animal Care Committee of Hebei Agricultural University (No. 14021).

### 2.2. DNA Extraction and PCR Amplification for Identification of SNPs

Genomic DNA was extracted from the liver tissues of adult cattle, placentas, and their corresponding mother’s whole blood and father’s sperm using the DNA Extraction Kit (Sangon Biotech, Shanghai, China). *MKRN3*, *MAGEL2* and *NDN* are three protein-coding genes with a single exon. SNP sites in the *MKRN3*, *MAGEL2* and *NDN* genes were identified by direct sequencing of the PCR products. Gene-specific primers were designed according to the mRNA sequences of the bovine *MKRN3* gene (XM_024982255.1), the *MAGEL2* gene (XM_002696471.5) and the *NDN* gene (NM_001014982.1). The primer sequences, the length of products and AT (annealing temperature) are shown in Table 1. 

PCR was performed in a 25-μL volume containing 1 μL of forward and reverse primers (10 μmol/L), 1 μL of genomic DNA template (100 ng), 12.5 μL of ES Tap Master Mix (CWBIO, Beijing, China) and 9.5 μL of ddH_2_O. The PCR conditions were as follows: 94 °C for 5 min, followed by 35 cycles of 94 °C for 30 s, AT for 30 s, 72 °C for 30 s and a final extension at 72 °C for 10 min. The amplified products were recovered and purified with a UNIQ-10 column DNA gel extraction kit (Sangon, Shanghai, China) and were sequenced directly using an ABI PRISM 3730 automated sequencer (Applied Biosystems, Massachusetts, MA, USA). An overlapping doublet peak in the sequencing chromatogram reveals the existence of a SNP. Heterozygous animals with identified SNPs were employed to analyze allelic expression.

### 2.3. Allelic Expression Analysis by RT-PCR

Total RNA was extracted from the tissues (including heart, liver, spleen, lung, kidney, muscle, fat, and brain) of heterozygous cows and heterozygous placentas using a TRIzol RNA extraction kit (Invitrogen, Massachusetts, MA, USA), and DNase-Ι treatment was performed to remove possible contamination of genomic DNA. The OD value was measured using an ND-1000 spectrophotometer (NanoDrop, Carlsbad, CA, USA), and a final concentration of 1000 ng/μL was prepared for reverse transcription (RT) experiments. The RT reaction was performed using oligo (dT) primers with a reverse transcription (RT) kit (Promega, Beijing, China) according to the manufacturer’s guidelines. The reaction volume and RT-PCR conditions were the same as those of PCR for SNP identification, except the template of genomic DNA was replaced with cDNA. The *MKRN3*, *MAGEL2* and *NDN* genes have single exons, and gene-specific primers for the amplification of genomic DNA were also used to amplify cDNA.

### 2.4. DNA Methylation Analysis by Bisulfite Sequencing PCR (BSP)

According to the locations of the PWS-IC DMR, the *MKRN3* DMR and the *NDN* DMR in humans, the CpG islands were predicted around the promoters of the *SNRPN*, *MKRN3* and *NDN* genes, and two methylation-specific primers for each CpG island were designed using the online website Methprimer 2.0 (http://www.urogene.org/methprimer2/, accessed on 1 November 2018). Genomic DNA (approximately 500 ng) extracted from tissues of heterozygous cattle, heterozygous placentas, sperm, and oocytes was sodium bisulfate treated with the EZ DNA Methylation-Gold^TM^ Kit (Zymo, Beijing, China). Nested PCR was performed to amplify the target products using FastPfu Fly DNA Polymerase with the treated DNA as template. The second-round PCR was performed using 1 μL of a tenfold dilution of the first-round PCR product as template, then the amplicons were purified and cloned into pMD19-T vectors (Takara, Beijing, China) for sequencing. For each product, at least 20 clones were sequenced. For each detected tissue, the percentage of mCpG(mCpG/CpG) in both parental strands was calculated. The sequences of the methylation-specific primers and AT are shown in Table 1.

## 3. Results

### 3.1. Monoallelic Expression of MKRN3, MAGEL2 and NDN in Bovine Tissues

To investigate the allelic expression of *MKRN3*, *MAGEL2* and *NDN* genes in bovine tissues, three SNPs, the c.1271T>G SNP (rs42331804) in the *MKRN3* sequence (XM_024982255.1), the c.457A>G SNP (rs211249225) in the *MAGEL2* sequence (XM_002696471.5) and the c.1336A>C SNP (rs468002089) in the *NDN* sequence (NM_001014982.1), were first detected by direct sequencing of PCR products. Heterozygous animals with these SNPs were used for allelic expression analysis of the *MKRN3*, *MAGEL2* and *NDN* genes (Figure 1). RT-PCR was performed on total RNA isolated from eight tissues (heart, liver, spleen, lung, kidney, muscle, fat and brain) of heterozygous animals. The amplified products of RT-PCR were sequenced directly. The sequencing chromatograms at the polymorphism sites indicated monoallelic expression of the *MKRN3*, *MAGEL2* and *NDN* genes in bovine heart, liver, spleen, lung, kidney, muscle, fat and brain tissues (Figure 1).

### 3.2. Paternally Expressed MKRN3, MAGEL2 and NDN in Bovine Placenta

The SNPs c.1271T>G (rs42331804) and c.1336A>C (rs468002089) were also evaluated to investigate the imprinted status of *MKRN3* and *NDN* in bovine placentas, respectively. A new SNP, c.2640A>G (rs110762305), was identified in the *MAGEL2* sequence(XM_002696471.5) and was used to analyze the imprinted expression of *MAGEL2* in placentas because no heterozygous placenta with the c.457A>G (rs211249225) SNP was found.

Based on SNP of c.1271T>G (rs42331804) in *MKRN3*, five heterozygous individuals were identified from 30 placentas. RT-PCR was also performed on total RNA isolated from heterozygous placentas. The amplified products of RT-PCR were the sequenced, and the results indicated that monoallelic (G) expression occurred in all five placentas. To determine which parental allele was expressed, the corresponding genomic DNA isolated from the maternal blood and paternal sperm samples was amplified. In three placentas (3–5, 1 = 18 and 1 = 12), the paternal and maternal genotypes were homozygous GG and heterozygous TG, respectively; thus, *MKRN3* exhibits paternal expression (allele G). For placentas 3501 and 3–16, both of the paternal and maternal genotypes were heterozygous TGs; therefore, we could not determine which parental allele was expressed (Figure 2A).

Four heterozygous placentas were found based on the c.2640A>G (rs110762305) SNP in the *MAGEL2* gene. Monoallelic (G) expression occurred in all four heterozygous placentas. Analysis of the parental genotypes revealed that the allele (G) was derived from the paternal allele (Figure 2B). In the three heterozygous placentas with the c.1336A>C (rs468002089) SNP of the *NDN* gene, the expressed allele (C) also came from the paternal allele (Figure 2C).

### 3.3. Identification of Bovine PWS-IC, MKRN3 and NDN DMR

In humans, three DMRs, PWS-IC, *MKRN3* DMR and *NDN* DMR, are located around the promoters of the *SNRPN*, *MKRN3* and *NDN* genes, respectively. To evaluate the role of DNA methylation in the regulation of monoallelic or imprinted expression of the bovine *MKRN3*, *MAGEL2* and *NDN* genes, the locations of these three bovine DMRs were first identified. Similar to what has been observed in humans, three CpG islands (CGIs) were identified in the promoter and exon 1 of the bovine *SNRPN* gene, the 5’UTR (5’ untranslated region) of the *MKRN3* gene and the promoter region of the *NDN* gene using CpG Plot analysis (http://www.ebi.ac.uk/Tools/emboss/cpgplot/, accessed on 1 November 2018). The locations of these three CGIs were of potential DMRs of PWS-IC, *MKRN3* and *NDN*. Methylation-specific primers were designed for bisulfite sequencing analysis of a 472-bp fragment containing 35 CpG dinucleotides in the PWS-IC CGI, a 406-bp fragment containing 25 CpGs in the *MKRN3* CGI and a 331-bp fragment containing 28 CpGs in the *NDN* CGI (Figure 3A). Because *SNRPN*, *MKRN3* and *NDN* genes exhibit similar paternal or monoallelic expression in bovine tissues, placental and two somatic tissues were selected to study their methylation.

The methylation status of PWS-IC CGI was tested in the brains, spleens and placentas (Figure 3B and Appendix A). A G/A polymorphism (rs441589013) was identified and used to discriminate the two parental alleles. As shown in Appendix A, there was a methylation level difference between the two parental alleles in the detected brains, spleens and placentas. The average methylation levels of the parental alleles were 88.0% (G allele) and 13.75% (A allele) in brains, 88.5% (G allele) and 19.2% (A allele) in spleens, and 82.4% (G allele) and 22.3% (A allele) in placentas. The difference in methylation level between parental alleles was more than 50% and exhibited a DMR-like methylation pattern; thus, it was named PWS-IC DMR. When examining the sperm and oocyte methylation levels (Figure 3B and Appendix A), unmethylation was observed in sperm across this region, and high hypermethylation was observed in the oocyte. These results suggest that the methylation pattern of PWS-IC DMR is established at the gamete stage and belongs to the primary regulatory DMR.

The methylation status of *MKRN3* CGI was analyzed in the brains, spleens and placentas (Figure 3C and Appendix A). A G/A SNP (rs433598400) distinguishes parental alleles into G and A alleles. The methylation pattern showed a DMR-like methylation pattern in all the detected tissues, and the methylation level of allele G was obviously lower than that of allele A (Appendix A). In *NDN* CGI, a G/T SNP (rs443578860) was used to distinguish parental alleles, and a methylation status similar to that of *MKRN3* CGI was observed in the detected brains, spleens and placentas (Figure 3D and Appendix A). The methylation level of the maternal allele of T was higher than that of the paternal allele of G (>50%). Therefore, the *MKRN3* CGI and the *NDN* CGI were named *MKRN3* DMR and *NDN* DMR, respectively. For *MKRN3* DMR and *NDN* DMR, hypomethylation was observed in both sperm and oocytes, indicating that the methylation patterns of *MKRN3* DMR and *NDN* DMR are established post fertilization and belong to the second regulatory DMR.

## 4. Discussion

Parental genomes are not functionally equivalent in embryonic development and body composition in mammalian species [17]. Genomic imprinting is a phenomenon of parent-of-origin effects(POE), which causes gene monoallelically expressed in paternal- or maternal-specific manner [18]. the parent-of-origin effects on quantitative trait loci (QTL) has been reported in livestock, such as cattle [19,20], pigs [21,22] and sheep [23,24]. Cattle, an economically important domestic farm animal species, serve as a potential model species in research on human preimplantation embryo development [25] and in determining the genetic etiology of sporadic human disorders [26]. To date, about 50 genes are experimentally identified to be imprinted in cattle (www.geneimprint.com). Most of these imprinted genes are validated by SNP-based method [27,28,29]. The high throughput sequencing technology provide new method for the discovery of new imprinted genes at the transcriptome level and has been performed in characterization of genomic imprinting in mice and humans [30,31,32,33]. Chen et al. assessed the imprinted gene expression in the bovine conceptus by the next generation sequencing technology and identified eight novel bovine imprinted genes and demonstrated that cis-eQTL effects can lead to the monoallelic expression of genes [34]. Imumorin et al. reported that quantitative trait loci with POE could affect seven growth and carcass traits in genome of Angus×Brahman cattle cross breds, and indicated that POE on quantitative traits is common in mammals, and the effects of imprinted gene on quantitative traits should be investigated [35].

*MKRN3*, *MAGEL2* and *NDN* are three protein-coding genes located in the human 15q11-13 imprinted region, and the absence of the paternal contribution of this chromosomal region could lead to human Prader-Willi syndrome, characterized by neurogenetic disorders [12,36,37]. The *MKRN3* gene, also known as *ZNF127*, is a protein called makorin ring finger protein 3. *MKRN3* serves as a repressor regulating puberty in humans [38,39]. The genetic alterations of *MKRN3* have been confirmed as the cause of central precocious puberty (CPP), including deleterious mutations in the coding region [38,39,40] and the promoter and 5′-UTR regulatory regions of the *MKRN3* gene [41,42]. *NDN* and *MAGEL2*, two members of the NDN/MAGEL2 gene family, encode necdin protein, which is concurrently inactivated in patients with PWS [43,44]. *MKRN3*, *MAGEL2* and *NDN* are candidate genes of PWS, but a deficiency of paternal deletion *MKRN3*, *MAGEL2* and *NDN* is not sufficient to result in PWS [45]. *NDN* gene could serve as a phylogenetic marker for studying the evolution and conservation in Bovidae species for its sufficient variation in gene sequence [46].

*NDN/Ndn* [12,44,47], *MAGEL2/Magel2* [36] and *MKRN3/Mkrn3* [13,48] were first identified to be paternally expressed in the central nervous system (CNS) and then confirmed to be imprinted in the E15.5 whole brains of mice [49]. *NDN and MAGEL2* are also confirmed to be imprinted in adult human tissues [33]. In this study, we showed that bovine *MKRN3*, *MAGEL2* and *NDN* genes are monoallelically expressed in adult somatic tissues, similar to their orthologs in humans and mice, although the tissue distribution of expression differs. In addition, *MKRN3* also exhibits paternally expressed genes in the rabbit brain [50].

Most well-known imprinted genes are highly expressed in pregnancy and term placental tissue and play critical roles in regulating human placental function and fetal development [51,52,53,54]. Some investigations of imprinted genes focus on placental tissue because placental tissue is not included in the Genotype-Tissue Expression (GTEx) project. The number of identified imprinted genes in the human placenta is consistent with the data on the mouse placenta [55]. *MKRN3* has been reported to be imprinted in the human placenta [56]. In this study, we showed that *MKRN3*, *MAGEL2* and *NDN* are paternally expressed in bovine placentas.

The parent-of-origin monoallelic expression of genes in an imprinted cluster is typically governed by DMRs inherited from the germline in the imprinting control region (ICR) [57,58]. In contrast to germline DMRs (gDMRs), differential DNA methylation in secondary DMRs (sDMRs) is acquired during embryonic development. The majority of sDMRs are usually regulated by a gDMR [59]. There are 24 identified imprinted gDMRs in the mouse genome: 21 maternal gDMRs are acquired during oogenesis, and three paternal gDMRs are acquired during spermatogenesis [58,60].

The ICR controlling genomic imprinting across the PWS/AS domain is composed of two parts: PWS-IC and AS-IC. In both mice and humans, PWS-IC is a maternal gDMR and is associated with the 5’ region of the *Snrpn/SNRPN* gene [61,62,63]. PWS-IC controls the paternal epigenotype on the paternally inherited chromosome [11,64]. The deletion of paternally inherited PWS-IC results in epigenetically silenced paternally expressed genes, such as *NDN*, *MAGEL2* and *MKRN3* [14,62,64,65,66,67,68]. These paternally expressed genes are also associated with a secondary differentially methylated region, which is acquired during embryonic development and exhibits paternally unmethylated and maternally hypermethylated regions [68,69,70].

In this study, we identified that the bovine primary DMR of PWS-IC was located in the promoter and the first exon of the bovine *SNRPN* gene, with hypomethylation in sperm and high hypermethylation in the oocyte. These results are consistent with the findings of previous studies by O’Doherty et al. [71] and Lucifero et al. [72]. The methylation of PWS-IC was consistent with the paternal or monoallelic expression of bovine *MKRN3*, *MAGEL2* and *NDN*, indicating that their expression was controlled by PWS-IC. For *MKRN3* and *NDN* gene, two somatic DMRs also identified to locate in the 5’UTR of the *MKRN3* gene and the promoter region of the *NDN* gene, respectively. Their locations and methylation patterns are similar to those in humans and mice; therefore, the imprinting regulation of this region appears to be conserved in humans, mice and cows.

## 5. Conclusions

The *MKRN3*, *MAGEL2* and *NDN* genes exhibit monoallelic or maternal imprinted expression in bovine somatic and placenta tissues, which was controlled by PWS-DMR located in the promoter and the first exon of the bovine *SNRPN* gene. The expression of *MKRN3* and *NDN* gene was also controlled by two somatic DMRs, *MKRN3* DMR and *NDN* DMR, respectively.

## Figures and Tables

**Figure 1 animals-11-01985-f001:**
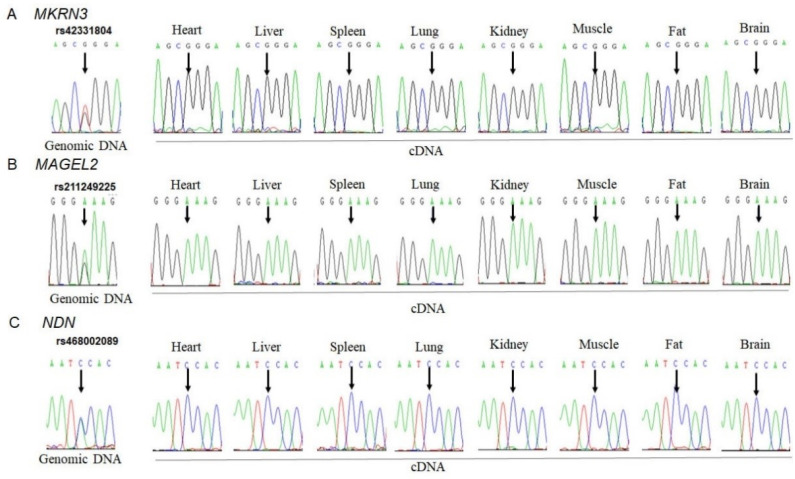
Monoallelic expression of *MKRN3*, *MAGEL2* and *NDN* in bovine tissues detected by RT-PCR. (**A**) Monoallelic expression of *MKRN3* in eight adult somatic tissues, including heart, liver, spleen, lung, kidney, muscle, fat and brain. The arrows indicate the SNP of rs42331804. (**B**) The SNP of rs211249225 in the *MAGEL2* gene and monoallelic expression of the bovine *MAGEL2* gene in eight detected tissues. (**C**) The SNP of rs468002089 in the *NDN* gene and monoallelic expression of the *NDN* gene.

**Figure 2 animals-11-01985-f002:**
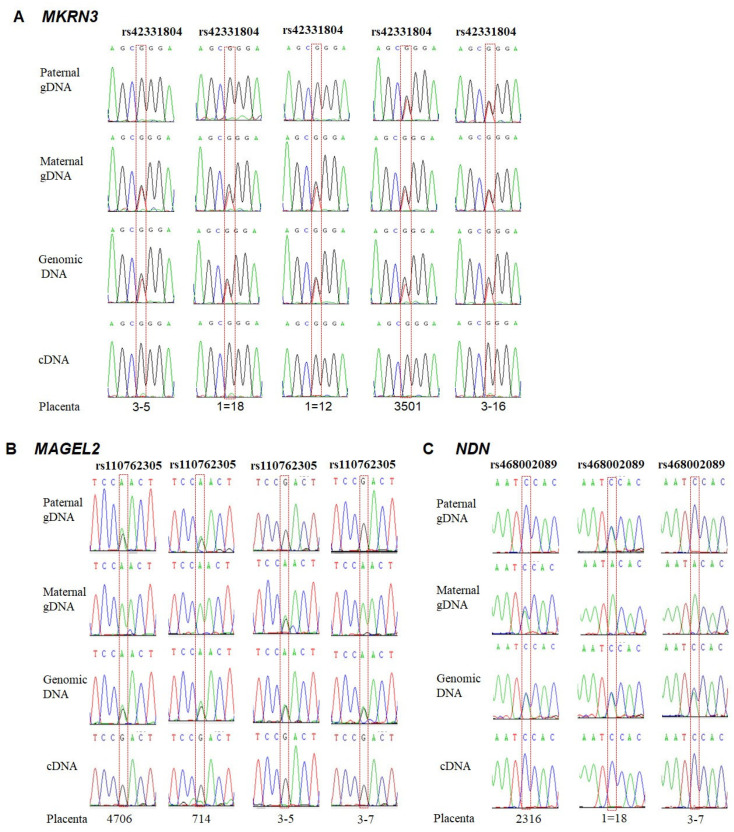
Paternal expression of *MKRN3*, *MAGEL2* and *NDN* in the bovine placenta. (**A**) Sequencing results of SNP (rs42331804) in the *MAGEL2* gene. Paternal expression was deduced by comparing genotypes of gDNA, cDNA and parental gDNA in five heterozygous placentas. (**B**) Paternal expression of the *MAGEL2* gene based on the sequencing results of the SNP rs110762305 in four heterozygous placentas. (**C**) Paternal expression of *NDN* deduced by sequencing results of SNP rs468002089 in three heterozygous placentas.

**Figure 3 animals-11-01985-f003:**
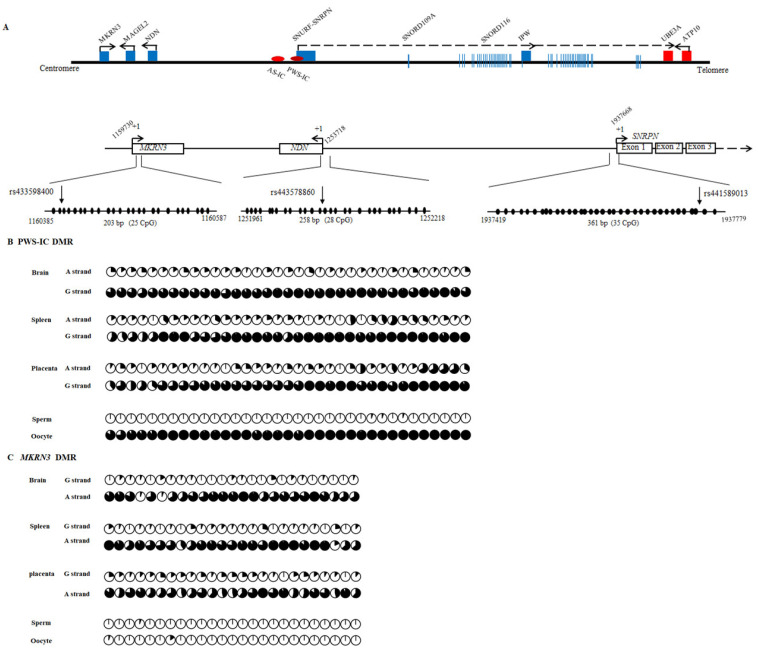
The structure of bovine PWS/AS imprinted region and the locations and methylation profiles of the PWS-IC, *MKRN3* and *NDN* DMRs in this region. (**A**) The structure of the PWS/AS imprinted region and the relative location of the PWS-IC, *MKRN3* and *NDN* DMRs in this locus. Maternally and paternally expressed genes are represented by red and blue squares, respectively. The transcription orientation is denoted with an arrow. The SNPs used to determine allele-specific methylation are shown with vertical arrows. (**B**) DNA methylation profiles of PWS-IC DMR in the brain, spleen, placenta, sperm and oocytes. (**C**) DNA methylation profiles of *MKRN3* DMR in the brain, spleen, placenta, sperm and oocytes. (**D**) DNA methylation profiles of *NDN* DMRs in the brain, spleen, placenta, sperm and oocytes. The allele origin of each clone was determined using SNPs in the heterozygous individuals. Methylation of each CpG site is presented with pie charts. The percent of methylated cytosines is represented in black, and the percent of unmethylated cytosines is represented in white at each site. The complete bisulfite maps of PWS-IC, *MKRN3* and *NDN* DMRs may be found in Appendix A.

**Table 1 animals-11-01985-t001:** Primers used in SNP identification, allelic expression and DNA methylation analysis.

Primer	Sequence (5′-3′)	AT (°C)	Product Size (bp)	Application
MKRN3-F MKRN3-R	CGCCCGTCACGTCTGA TACAGCTCTGCCCACGAAAG	55	568	Identification of SNP rs42331804 and allelic expression analysis of*MKRN3*
MAGEL2-F1 MAGEL2-R1	GAAAAACTTGCCTACCACATC CCACATCCCTGAGCAAGAGTA	56	696	Identification of SNP rs211249225 and allelic expression analysis of*MAGEL2*in tissues
MAGEL2-F2 MAGEL2-R2	CGTAGGCATTCTCTTCTCTC AACCTGTGACTGGATCTGC	56	1045	Identification of SNP rs110762305 and allelic expression analysis of*MAGEL2*in placenta
NDN-F NDN-R	AGAAACACTCCACCTTCG CTACCCCAATACACAGCC	55	644	Identification of SNP rs468002089 and allelic expression analysis of *NDN*
PWS-IC-F1 PWS-IC-R1	AAGGAAATTGATAGTAAGTATATTAGAGT AACCCAAATCCCCAATAAA	59	791	PWS-IC DMR identification and analysis
PWS-IC-F2 PWS-IC-R2	GTTATTAGTGGAAAGTTTGAGGAAA ACCACACGACTAACCTAACCC	59	472	
MKRN3 DMR-F1 MKRN3 DMR-R1	TGTAAGAATTATTAGAAAATAAAGAGTAGA CCCAATCCCTACTTCCTATACCTA	54	541	MKRN3 DMR identification and analysis
MKRN3 DMR-F2 MKRN3 DMR-R2	TATATAGATATAAAATGAAGTGAATAAAG CCTACTTCCTTCTCTAAACAAA	52	406	
NDN DMR-F1 NDN DMR-R1	AGTTTTAATAGGACGTTTGGGGAGG AATCTCCCTCTTCGCCTAAAACCTA	59	526	NDN DMR identification and analysis
NDN DMR-F2 NDN DMR-R2	GGGAGTGATTATTGAGGTTTA ACAAAATTAAAATTACACACATCCTTACT	59	331	

## Data Availability

The data reported in this study is available on request from the corresponding author.

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
