# Peer review of "Conservation of Imprinting and Methylation of MKRN3, MAGEL2 and NDN Genes in Cattle"

_animals, 2021, doi:10.3390/ani11071985_

Round 1
Reviewer 1 Report
The manuscript “Conservation of imprinting and methylation of MKRN3, 1 MAGEL2 and NDN genes in cattle” contains relevant information that can contribute to the knowledge of the regulatory mechanisms of imprinting genes by comparison between species but some minor change should be made and a grammar checking should be made before publication.
The sentence between lines 13-14 should be included the words “knowledge of “. In the manuscript, the authors are describing mechanisms and genes that are already occurring. In my point of view, the manuscript is helping to increase the knowledge of genes under this effect. As result, they are increasing the list of already know imprinting genes.
The authors should also check the redaction. Some mistakes can be observed when you read the document. For example, in line 30 (wasa, should be was a), in line 31 (weresecondary should be were secondary), in line 163 there is a “the” that do not fit in this sentence, in line 173 ( inthe, should be in the), and in lines in 278-279 (“were then were confirmed”, it is not clear what they mean with that), between others.
In material and methods section can be useful to have more information about the population used in the experiment. For example, in line 74 is mentioned that samples were taken from 34 Holstein cows, and in line 76 the authors mention that placenta was taken from local cattle. Those local cattle are local breeds or are they Holstein? In addition, what about the father information, are the fathers different for each placenta or is the same bull for all of it? How many bulls did they evaluate in this experiment? This is not clear in the manuscript.
Author Response
This work could advance the knowledge of genomic imprinting by increase the number of imprinted genes in bovine and also facilitate future studies to detect the physiological roles of MKRN3, MAGEL2 and NDN genes.
Those local cattle are Holstein and the fathers are different for each placenta, almost 34 Holstein cows we evaluated in this experiment.
The manuscript “Conservation of imprinting and methylation of MKRN3, MAGEL2 and NDN genes in cattle” contains relevant information that can contribute to the knowledge of the regulatory mechanisms of imprinting genes by comparison between species but some minor change should be made and a grammar checking should be made before publication.
Re : We are very grateful to your comments and suggestions. We have made a careful grammar checking and corrected errors in our manuscript.
The sentence between lines 13-14 should be included the words “knowledge of “. In the manuscript, the authors are describing mechanisms and genes that are already occurring. In my point of view, the manuscript is helping to increase the knowledge of genes under this effect. As result, they are increasing the list of already know imprinting genes.
Re : Thanks for your suggestions. The words “knowledge of “ has been added and this sentence has changed to be “This work could advance the genomic imprinting by increasing the knowledge of imprinted genes in bovine and also facilitate future studies to detect the physiological roles of MKRN3, MAGEL2 and NDN genes.”.
The authors should also check the redaction. Some mistakes can be observed when you read the document. For example, in line 30 (wasa, should be was a), in line 31 (weresecondary should be were secondary), in line 163 there is a “the” that do not fit in this sentence, in line 173 ( inthe, should be in the), and in lines in 278-279 (“were then were confirmed”, it is not clear what they mean with that), between others.
Re:We are very sorry for our careless and we have changed these mistakes.
In material and methods section can be useful to have more information about the population used in the experiment. For example, in line 74 is mentioned that samples were taken from 34 Holstein cows, and in line 76 the authors mention that placenta was taken from local cattle. Those local cattle are local breeds or are they Holstein? In addition, what about the father information, are the fathers different for each placenta or is the same bull for all of it? How many bulls did they evaluate in this experiment? This is not clear in the manuscript.
Re : Thanks for your suggestions. The more information about the population used in the experiment has been added in our revised manuscript.
Reviewer 2 Report
This manuscript reports on the imprinted status of the MKRN3, MAGEL2, and NDN genes in the bovine. These three genes are maternally imprinted, paternally expressed as part of a imprinted gene cluster at the Prader-Willi and Angelman syndromes locus at 15q11-q13. PWS is contiguous gene disorder, and the three genes examined here are among several protein coding genes and ncRNAs implicated in PWS traits. The evolutionary assembly of the locus has been examined, but patterns of imprinting have been primarily tested only in humans and rodents.
To the beat of my knowledge, the current work is novel in providing the first demonstration that MKRN3, MAGEL2, and NDN are imprinted in the bovine. Briefly, the approach was to identify offspring heterozygous for SNPs in these genes, and subsequently identify whether the expressed polymorphism was biallelically, maternally, or paternally expressed. At this imprinted cluster, these genes are hypermethylated on the maternal allele. The authors have also used bisulfite genomic sequencing to confirm that the bovine homologs are similarly hypermethylated.
The major strength of this manuscript is that the authors have provided a careful and thorough analysis. They examine gene expression not only in brain but also seven additional tissues. They also show that these genes exhibit imprinted gene expression in the placenta, a tissue that has been hard to pin down in the human.
I have the following concerns about the manuscript.
- These genes are among a cluster of paternally expressed genes including SNRPN. There is some prior work demonstrating that SNRPN is paternally expressed in the bovine, yet surprisingly this previous work is not mentioned, nor is work demonstrating a primary DMR at SNRPN promoter-exon 1 region. See [Smith Reprod Dom Anim 47, 107–114 (2012); doi: 10.1111/j.1439-0531.2012.02063.x] [O’Doherty Biology Of Reproduction (2012) 86(3):67, 1–10] [Lucifero Biology Of Reproduction 75, 531–538 (2006)]
- Line 26. The abstract says “we found that the MKRN3, MAGEL2 and NDN genes exhibit monoallelic expression in bovine somatic tissues and the maternal allele expressed in the bovine placenta. “ This statement is in direct opposition to the results.
- Minor issue: Line 303: The deletion of paternally inherited PWS-IC results in epigenetically silenced paternally expressed 304 genes, such as NDN, MAGEL2 and MKRN3 [62,64–68]. The paternally expressed gene is also associated with a differentially methylated region around the gene promoter region, which is acquired during embryonic development and exhibits paternally unmethylated and maternally hypermethylated regions [69–71]. For readers less familiar with imprinting, this sentence might be easier to understand if it started with “These paternally expressed genes are also associated with a secondary differentially methylated region.
Author Response
SNRPN alleles were unmethylated in sperm, methylated in oocytes, and approximately 50% methylated in somatic samples [Lucifero Biology Of Reproduction 75, 531–538 (2006)] and there is a generalized hypomethylation of the imprinted allele and the biallelic expression of embryos produced by SCNT when compared to the methylation patterns observed in vivo (artificially inseminated) for SNRPN [Smith Reprod Dom Anim 47, 107–114 (2012); doi: 10.1111/j.1439-0531.2012.02063.x]
Placenta is a special organ, which has an important influence on the growth and development of fetus, many imprinted genes are placenta specific, and many imprinted genes are not imprinted in placenta, so we found that the MKRN3, MAGEL2 and NDN genes exhibit monoallelic expression in bovine somatic tissues and the maternal allele expressed in the bovine placenta.
I have the following concerns about the manuscript.
- These genes are among a cluster of paternally expressed genes including SNRPN. There is some prior work demonstrating that SNRPN is paternally expressed in the bovine, yet surprisingly this previous work is not mentioned, nor is work demonstrating a primary DMR at SNRPN promoter-exon 1 region. See [Smith Reprod Dom Anim 47, 107–114 (2012); doi: 10.1111/j.1439-0531.2012.02063.x] [O’Doherty Biology Of Reproduction (2012) 86(3):67, 1–10] [Lucifero Biology Of Reproduction 75, 531–538 (2006)]
Re:We are very sorry for our careless. We have added these studies in the discussion section [reference 72,73].
- Line 26. The abstract says “we found that the MKRN3, MAGEL2 and NDN genes exhibit monoallelic expression in bovine somatic tissues and the maternal allele expressed in the bovine placenta. “ This statement is in direct opposition to the results.
Re:We are very sorry for our careless. We have changed the error statement “maternal allele expressed” to “paternal allele expressed”.
- Minor issue: Line 303: The deletion of paternally inherited PWS-IC results in epigenetically silenced paternally expressed 304 genes, such as NDN, MAGEL2 and MKRN3 [62,64–68]. The paternally expressed gene is also associated with a differentially methylated region around the gene promoter region, which is acquired during embryonic development and exhibits paternally unmethylated and maternally hypermethylated regions [69–71].For readers less familiar with imprinting, this sentence might be easier to understand if it started with “These paternally expressed genes are also associated with a secondary differentially methylated region.
Re:We are truly thankful for your suggestions. We have inserted this sentence in our manuscript.
Reviewer 3 Report
Few Minor Comments:
Line 8: Replace “farm animals” to “farm animal”
line 30: replace “wasa” to “was a” or even better “is a”
line 31: replace ‘weresecondary’ to “were secondary” or even better “are secondary”
Line 68: Does the imprinting status of the cow here means you only sampled female cattle?
Line 230: the resolution of Figure 3, in particular for the “A” part should be improved
Line 279: remove “were”
The novelty of the study, and the conflicting points
The authors investigated the regulatory mechanism (Methylation pattern) of the expression of three clustered imprinted genes (MKRN3, MAGEL2 and NDN) located on Chromosome 21, through identifying the methylated regions in CpG islands. They found the PWS-IC DMR is the primary DMR inherited from gametes, but NDN and MKRN3 DMS are secondary DMRs and come into play after fertilization. These results bring new information for the bovine research community and in particular are important for epigenetics studies and bovine selective breeding in future. In addition, they show the NDN gene exhibit monoallelic (paternal) expression in bovine somatic and placenta tissues. However, the monoallelic expression of the genes MKRN3 and MAGEL2 is not new to this study as described below:
The author claimed that the genes MKRN3, MAGEL2 and NDN are maternally imprinted in human, however their imprinting status and epigenetic mechanism in cattle is still unknown, and therefore this study advances the genomic imprinting in cattle by adding more imprinted genes into the list. This statement is in contrast to the finding by Chen et al. (2016) reporting that MKRN3 and MAGEL2 are imprinted in bovine. Chen et al. (2016) validated the allele specific expression (paternal expression) of MKRN3 in the bovine brain tissue and suggested that this gene is indeed imprinted. In addition, they confirmed the imprinted expression (paternal expression) of the previously reported imprinted gene MAGEL2 in bovine (https://www.geneimprint.com/site/genes-by-species.Bos+taurus) through RNA-seq analysis in brain, kidney, liver, skeletal muscle, and placenta. Therefore, only the imprinted pattern in NDN gene in bovine is new. However, the expression of these three genes have been investigated in more tissues compare to the previous studies.
The paternal expression of MKRN3:
The authors show the monoallelic expression of the gene MKRN3 (SNP c.1271T>G, rs42331804) in eight adult cattle tissues. In addition, they investigated the paternal expression of MKRN3 in placental of three out of five heterozygous individuals for the SNP c.1271T>G (rs42331804). In both adult tissues, and placenta the allele G is expressed. Considering the paternal expression is only confirmed in 60% of the placenta samples, the expression of “G” allele could be the result of cis-eQTL leading to allele specific gene expression rather than imprinting. An example of such cases is the expression of gene LOC101905472 in bovine is sequence-dependent and the allele “C” is always expressed regardless of the parental origin.
In regard to the results of this study, the paternal expression of MKRN3 in cattle brain tissues has already been confirmed (Chen et al. 2016), and it’s not the effect of cis-eQTL. Therefore, the results are valid but should either refer to the previous finding as a confirmation for their results, or use different mating structure crosses to confirm their own results prior to assuming any monoallelically expressed genes are imprinted.
Author Response
Line 8: Replace “farm animals” to “farm animal” re: change for farm animal
line 30: replace “wasa” to “was a” or even better “is a” re: change for is a
line 31: replace ‘weresecondary’ to “were secondary” or even better “are secondary” re: change for are secondary
Line 68: Does the imprinting status of the cow here means you only sampled female cattle? re: no, we sampled almost all cattles
Line 230: the resolution of Figure 3, in particular for the “A” part should be improved re: Improved
Line 279: remove “were” re: done
Re:We are very sorry for our careless and thanks for your suggestions. We have changed these mistakes.
the others: ok, you are right.
get it
The novelty of the study, and the conflicting points
The authors investigated the regulatory mechanism (Methylation pattern) of the expression of three clustered imprinted genes (MKRN3, MAGEL2 and NDN) located on Chromosome 21, through identifying the methylated regions in CpG islands. They found the PWS-IC DMR is the primary DMR inherited from gametes, but NDN and MKRN3 DMS are secondary DMRs and come into play after fertilization. These results bring new information for the bovine research community and in particular are important for epigenetics studies and bovine selective breeding in future. In addition, they show the NDN gene exhibit monoallelic (paternal) expression in bovine somatic and placenta tissues. However, the monoallelic expression of the genes MKRN3 and MAGEL2 is not new to this study as described below:
The author claimed that the genes MKRN3, MAGEL2 and NDN are maternally imprinted in human, however their imprinting status and epigenetic mechanism in cattle is still unknown, and therefore this study advances the genomic imprinting in cattle by adding more imprinted genes into the list. This statement is in contrast to the finding by Chen et al. (2016) reporting that MKRN3 and MAGEL2 are imprinted in bovine. Chen et al. (2016) validated the allele specific expression (paternal expression) of MKRN3 in the bovine brain tissue and suggested that this gene is indeed imprinted. In addition, they confirmed the imprinted expression (paternal expression) of the previously reported imprinted gene MAGEL2 in bovine (https://www.geneimprint.com/site/genes-by-species.Bos+taurus) through RNA-seq analysis in brain, kidney, liver, skeletal muscle, and placenta. Therefore, only the imprinted pattern in NDN gene in bovine is new. However, the expression of these three genes have been investigated in more tissues compare to the previous studies.
The paternal expression of MKRN3:
The authors show the monoallelic expression of the gene MKRN3 (SNP c.1271T>G, rs42331804) in eight adult cattle tissues. In addition, they investigated the paternal expression of MKRN3 in placental of three out of five heterozygous individuals for the SNP c.1271T>G (rs42331804). In both adult tissues, and placenta the allele G is expressed. Considering the paternal expression is only confirmed in 60% of the placenta samples, the expression of “G” allele could be the result of cis-eQTL leading to allele specific gene expression rather than imprinting. An example of such cases is the expression of gene LOC101905472 in bovine is sequence-dependent and the allele “C” is always expressed regardless of the parental origin.
In regard to the results of this study, the paternal expression of MKRN3 in cattle brain tissues has already been confirmed (Chen et al. 2016), and it’s not the effect of cis-eQTL. Therefore, the results are valid but should either refer to the previous finding as a confirmation for their results, or use different mating structure crosses to confirm their own results prior to assuming any monoallelically expressed genes are imprinted.
Re:We are truly thankful for your suggestions.We have studied the document (Chen et al. , 2016) carefully and improved our understanding on genome imprinting.
We have deleted the statement of “ their imprinting status and epigenetic mechanism in cattle is still unknown”, and changed “ this study advances the genomic imprinting in cattle by adding more imprinted genes into the list” to “this work could advance the genomic imprinting by increasing the knowledge of imprinted genes in bovine”.
Reviewer 4 Report
Authors investigated imprinting of a cluster of genes on bovine chromosome 21 (human 15, mouse 7c) and found consistent monoallelic expression across the 3 species. Furthermore they identified methylation patterns that were consistent with methylation driven imprinting. Conclusions are well supported by the data. Great work! Other than a few typos of minor importance I found no problems with the manuscript.
Author Response
Thank you, best wish.
Thanks for your review on our manuscript, we have corrected typos in the text.